# Deep learning classification of shoulder fractures on plain radiographs of the humerus, scapula and clavicle

**Martin Magnéli* ⃝*, Petter Ling, Jacob Gislén, Johan Fagrell, Yilmaz Demir, Erica Domeij Arverud, Kristofer Hallberg ⃝, Björn Salomonsson ⃝, Max Gordon ⃝**

Department of Clinical Sciences at Danderyd Hospital, Karolinska Institutet, Stockholm, Sweden

* martin.magneli@ki.se

**Data Availability Statement:** The instructions how to get access to the dataset are available at the dataset page at AIDA: https://datahub.aida.scilifelab.se/10.23698/aida/sr2023.

## Abstract

In this study, we present a deep learning model for fracture classification on shoulder radiographs using a convolutional neural network (CNN). The primary aim was to evaluate the classification performance of the CNN for proximal humeral fractures (PHF) based on the AO/OTA classification system. Secondary objectives included evaluating the model's performance for diaphyseal humerus, clavicle, and scapula fractures. The training dataset consisted of 6,172 examinations, including 2–7 radiographs per examination. The overall area under the curve (AUC) for fracture classification was 0.89, indicating good performance. For PHF classification, 12 out of 16 classes achieved an AUC of 0.90 or greater. Additionally, the CNN model had excellent overall AUC for diaphyseal humerus fractures (0.97), clavicle fractures (0.96), and good AUC for scapula fractures (0.87). Despite the limitations of the study, such as the reliance on ground truth labels provided by students with limited radiographic assessment experience, our findings are in concordance with previous studies, further consolidating CNN as potent fracture classifiers in plain radiographs. The inclusion of multiple radiographs with different views from each examination, as well as the generally unselected nature of the sample, contributed to the overall generalizability of the study. This is the fifth study published by our group on AI in orthopaedic radiographs, which has consistently shown promising results. The next challenge for the orthopaedic research community will be to transfer these results from the research setting into clinical practice. External validation of the CNN model should be conducted in the future before it is considered for use in a clinical setting.

## Introduction

Shoulder fractures include fractures of the proximal humerus, clavicle, and scapula. Among these, proximal humerus fractures (PHF) are some of the most common fractures in the elderly population. PHF account for approximately 5% of all fractures and occur most commonly in women [1]. Fractures are often caused by minimal trauma, such as a fall from standing height or less [2,3]. PHF can be divided into fractures through the tuberosities, metaphysis,

**Funding:** The authors received no specific funding for this work.

**Competing interests:** The authors have declared that no competing interests exist.

surgical, and anatomical neck, in combination with a myriad of fragments. These range from simple extraarticular, unifocal fractures to complex articular, multifocal, and multifragmentary patterns that engage the entirety of the humeral head and metaphysis [4].

Classification of PHF was pioneered by Neer et al., who introduced the PHF 4-part classification based on fracture displacement and fragmentation patterns in 1970 [5,6]. The AO Foundation/Orthopaedic Trauma Association (AO/OTA) later introduced the AO/OTA Fracture and Dislocation Classification Compendium, most recently revised in 2018 [4]. However, there is no consensus among clinicians and researchers regarding which system is superior, and both systems are widely used and accepted in the orthopaedic community [7]. Interobserver agreement (IOA) in PHF classification can vary substantially depending on the skill level of the individual observer, and comparative studies of the Neer and AO/OTA PHF radiographic classifications suggest that IOA in PHF classification is fair to moderate at best, regardless of which classification is used [8–10].

Fractures of the clavicle and scapula are less common than humerus fractures, with clavicle fractures constituting 2.6–4% of fractures in adults and scapula fractures accounting for less than 1% of all fractures [11,12].

In recent years, neural network image classifiers have been established as an efficient model for data analysis in orthopaedic research [13]. Convolutional neural networks (CNN) have previously proven effective in detecting and classifying fractures in several anatomical locations. Chung et al. demonstrated the potential of a CNN in identifying and distinguishing PHF in plain anteroposterior (AP) radiographs using the Neer classification. When classifying complex fractures, their CNN performed better than experienced orthopaedic surgeons [14]. These findings suggest that CNN can detect and classify fractures with accuracy approximating and even surpassing that of human ability.

The aim of this study was to train and evaluate a CNN model for AO/OTA classification of shoulder fractures.

## Materials and methods

### Study design and sample

A total of 7,189 plain radiographic shoulder examinations, conducted between 2002 and 2016, were extracted from Danderyd Hospital Picture Archiving and Communication System (PACS). Examinations were conducted based on all standard indications for routine shoulder radiography at Danderyd Hospital, using standard pathology-specific protocols for radiographic shoulder assessment, where each examination consisted of 2–7 radiographs. Examinations were anonymized during the extraction process and were void of all patient data. Ethical permit was granted by the Stockholm Ethical Review Board, Sweden. Dnr: 2014/453-31/3. The Stockholm Ethical Review Board waived the need for informed consent for this study. In this study, shoulder fractures were defined as fractures of the humerus, scapula, or clavicle; however, humerus fractures were limited to proximal and diaphyseal fractures.

### Datasets

The study sample (n = 7,189) was divided into a training (n = 6,221), validation (n = 562), and test dataset (n = 406). No patient overlap was present among the datasets. Examinations were extracted based on radiologist reports indicating radiographic examinations of the shoulder. Specific projections were not considered a criterion for sample extraction. After reviewing initial network classification performance, classes with poor performance were identified. To improve performance in these classes, we used active learning by increasing the number of examinations including the specific fracture class. The PACS database was scanned for

radiologist reports containing wordings suggesting the selected fracture types. This introduced possible selection bias and was deemed acceptable to increase prediction precision for all classes.

## Labelling of radiographic examinations

The extracted radiographic examinations were uploaded to an in-house developed online labelling platform and were labelled with respect to fractures. The training and validation datasets were labelled by three 4th-year medical students. Radiologist's reports were attached to each examination when available, complementing the students in visual assessment. The labels were considered ground truth. Particularly ambiguous or difficult cases were revisited and re-audited by the medical students, often with senior surgeon supervision. The test set was labelled by four senior shoulder surgeons.

## Fracture labels

Fractures in the proximal and diaphyseal humerus, clavicle, and scapula were labelled according to the AO/OTA classification. Radiographs containing more than one characteristic fracture were labelled accordingly, except when two or more groups or subgroups within the same type occurred simultaneously, as the hierarchical structure of the labelling platform did not allow for multiple classes within the same principal fracture type. Fracture labels were applied in a hierarchical manner, with the option of not assigning groups and/or subgroups that could not be determined by the observer. However, the principal fracture type was registered and included in the study sample regardless.

## Definitions

**Proximal humerus.** Proximal humerus fractures are classified into 3 types, 6 groups, and 12 subgroups (Table 1). 'Class' was the compound term used for a specific group and/or subgroup and was addressed with the respective fracture code A1.1 for the greater tuberosity fracture. Qualifications and universal modifiers were applied when appropriate.

**Diaphyseal humerus.** Diaphyseal humerus fractures are classified into 3 types and 7 groups.

**Clavicle.** Clavicle fractures are classified into 3 locations and 9 types. In this study, only two locations, diaphyseal and lateral fractures, were used.

Scapula. Scapula fractures are classified into 3 locations, 8 types, and 5 groups. Process fractures (location A) were included as a group and not classified according to type. Body fractures (location B) were included and classified into types.

## Training the CNN and evaluating CNN classification performance

A modified CNN of the ResNet architecture was used with a total of 35 convolutional layers, with batch normalization for each convolutional layer and an adaptive max pool [15]. The network was randomly initialized and trained using stochastic gradient descent. The 6,172 student-labelled examinations were used as training examples in CNN training and were considered ground truth in this setting. The images in the training dataset were separately processed by the CNN for 80 epochs (rounds). Images were scaled down from original size to 256x256 pixels to fit the predefined image framework. The images were additionally randomly cropped, rotated, and inverted. During the training, the model was evaluated using the validation dataset, comprising 562 examinations. The final, adjusted model was evaluated using the test dataset, comprising 406 examinations.

**Table 1.**

| Proximal humerus | | | Clavicle | | |
|---|---|---|---|---|---|
| Type A: Extraarticular, unifocal, 2-part fractures. | | | Location 2: Diaphyseal segment. | | |
| | Group A1: Tuberosity fracture: | | | Type 2A: Simple fracture | |
| | | Subgroup A1.1: Greater tuberosity fracture | | Type 2B: Wedge fracture | |
| | | Subgroup A1.2: Lesser tuberosity fracture. | | Type 2C: Multifragmentary fracture | |
| | Group A2: Surgical neck fractures. | | Location 3: Lateral segment | | |
| | | Subgroup A2.1: Simple fracture. | | Type 3A: Extraarticular fracture | |
| | | Subgroup A2.2: Wedge fracture. | | Type 3B: Partial articular fracture | |
| | | Subgroup A2.3: Multifragmentary fracture. | | Type 3C: Complete articular fracture | |
| | Group A3: Extraarticular, vertical fracture. | | | | |
| Type B: Extraarticular, bifocal, 3-part fractures. | | | Scapula | | |
| | Group B1: Surgical neck fractures: | | Location A: Process | | |
| | | Subgroup B1.1: With greater tubercle fracture. | | Type A1: Coracoid fracture | |
| | | Subgroup B1.2: With lesser tubercle fracture. | | Type A2: Acromion fracture | |
| Type C: Articular or 4-fragment fractures. | | | | Type A3: Spine fracture | |
| | Group C1: Anatomical neck fractures: | | Location B: Body | | |
| | | Subgroup C1.1: Valgus impacted fracture. | | Type B1: Fracture exits the body at 2 or less points | |
| | | Subgroup C1.3: Isolated anatomical neck fracture. | | Type B2: Fracture exits the body at 3 or more points | |
| | Group C3: anatomical neck fracture associated with metaphyseal fractures: | | Location F: Glenoid fossa | | |
| | | C3.1: With a multifragmentary metaphyseal segment with intact articular surface. | | Type F0: Through the extraarticular subchondral bone of the glenoid fossa (glenoid neck) | |
| | | C3.2: With a multifragmentary metaphyseal segment with articular fracture. | | Type F1: Simple fracture | |
| | | C3.3: With a multifragmentary metaphyseal fracture, with diaphyseal extension and articular fracture. | | | Group F1.1: Anterior rim fracture |
| | | | | | Group F1.2: Posterior rim fracture |
| Diaphyseal humerus | | | | | Group F1.3: Transverse or short oblique fracture |
| Type A: Simple fractures. | | | | Type F2: Multifragmentary (three or more fracture lines) | |
| | Group A1: Spiral fracture. | | | | Group F2.1: Glenoid fossa fracture |
| | Group A2: Oblique fracture (>30˚). | | | | Group F2.2: Central fracture dislocation |
| | Group A3: Transverse fracture (<30˚). | | | | |
| Type B: Wedge fractures. | | | | | |
| | Group B2: Intact wedge fracture. | | | | |
| | Group B3: Fragmentary wedge fracture. | | | | |
| Type C: Multifragmentary fractures. | | | | | |
| | Group C2: Intact segmental fracture. | | | | |
| | Group C3: Fragmentary segmental fracture. | | | | |

## Statistics

Classification performance of the CNN model was evaluated using sensitivity, specificity, Area Under the Receiver Operating Characteristics (ROC) Curve (AUC), and Youden's index (J).

AUC is a value between 0 and 1. For this study, we chose to define AUC between 0.7 and 0.8 as "acceptable", AUC between 0.8 and 0.9 as "excellent", and AUC 0.9 or higher as "outstanding" based on an article by Mandrekar on diagnostic test assessment [16]. J is defined as $J = sensitivity+specificity-1$ and describes the maximum potential accuracy of a diagnostic test. All statistical analyses were performed with R, using publicly available MLmetrics packages and OptimalCutpoints for sensitivity, specificity, and J.

## Results

A total of 6,783 radiographic examinations were included and divided into a training dataset (n = 6,221), a validation dataset (n = 562), and a test dataset (n = 406). The numbers of the different types of shoulder fractures and their distribution in the different datasets are displayed in Fig 1.

### Proximal humerus fractures

**Distribution of fractures in the training data.** Most fractures belonged to extraarticular, unifocal, 2-part fracture (type A, n = 465) followed by extraarticular, bifocal, 3-part fracture (type B, n = 60) and articular or 4-part fracture (type C, n = 47). All 6 groups and 11/12 subgroups were represented in the training data. Tuberosity fracture (A1) was the most common group (n = 374), and isolated greater tuberosity fracture (A1.1) was the most common subgroup (n = 370). Surgical neck fracture with lesser tubercle fracture (B1.2) was the only subgroup not represented in the training data.

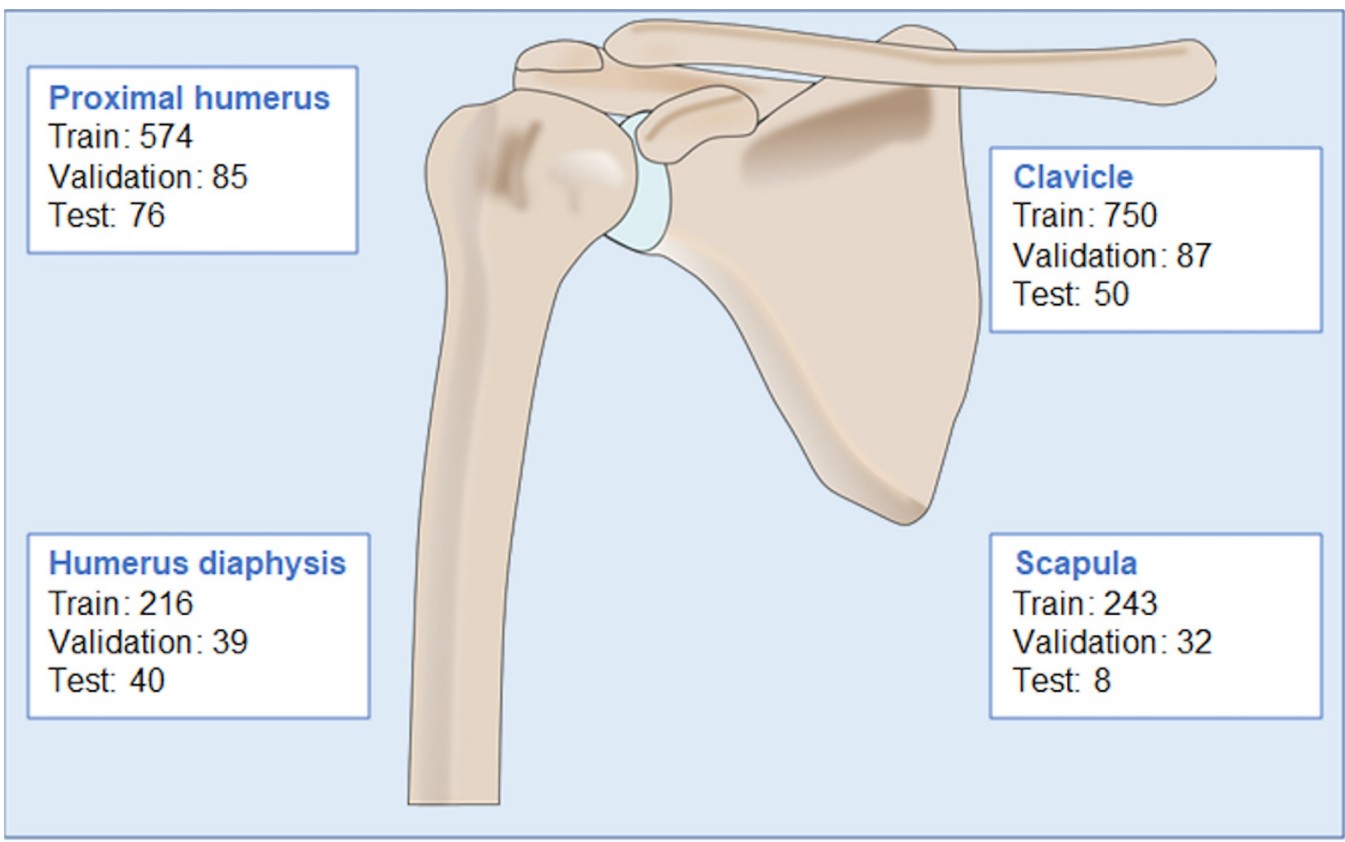

**Fig 1. Distribution of different fractures in the datasets.**

## Overall model performance

The CNN exhibited excellent overall PHF classification performance with AUC 0.92 (95% CI 0.88 to 0.95) for PHF. It classified 77 PHF with 83% sensitivity, 89% specificity, and J 0.79. A total of 329 examinations were classified as no PHF. The AUC was high (0.9) for all three fracture types. The most accurate class-specific predictions were found in the multifragmentary surgical neck fracture (subgroup A2.3) class with AUC 1.0 (95% CI 0.99 to 1.00) and J 0.99. The least accurate class-specific predictions were in the A2.1 class, with mean AUC 0.73 (95% CI 0.29 to 1.0), J 0.61. The predictive accuracy in all classes is displayed in Table 2.

A total of 406 examinations were used in the test set to evaluate the CNN classification performance, with 10 out of 13 PHF classes represented. Model performance for the different anatomical areas is summarized in Fig 2.

**Extraarticular, unifocal, 2-part fractures (type A).** The CNN detected and classified 60 type A fractures in the test set with high predictive accuracy of AUC 0.90 (95% CI 0.88 to 0.94), 76% sensitivity, 89% specificity, and J 0.66 (Table 2). The majority (47) were greater tuberosity fractures (subgroup A1.1). Predictive accuracy in this class was high with AUC 0.91 (95% CI 0.86 to 0.95). Isolated lesser tubercle fracture (subgroup A1.2) was not represented in the test data.

**Extraarticular, bifocal, 3-part fracture (type B).** A total of 5 fractures type B were detected with 100% sensitivity, 73% specificity, and J 0.73. Classification accuracy was high with AUC 0.90 (95% CI 0.80 to 1.00). The most accurate predictions were yielded in the surgical neck fracture with greater tuberosity fracture (B1.1) with AUC 0.95 (95% CI 0.90–1.00). Surgical neck fracture with lesser tuberosity fracture (subgroup B1.2) was not represented in the test data.

**Articular or 4-part fracture (type C).** 12 type C fractures were detected with 92% sensitivity, 81% specificity, and J 0.73. Predictive accuracy was good, with AUC 0.90 (95% CI 0.83 to 0.97). The most accurate type C classification predictions were found in the anatomical neck fracture with a multi-fragmentary metaphyseal segment with articular fracture class

**Table 2. Proximal humerus.**

|  | Cases (n = 406) | Measurements | | | |
|---|---|---|---|---|---|
|  |  | Sensitivity (%) | Specificity (%) | Youden's J | AUC (95% CI) |
| Type A |  |  |  |  |  |
| A | 60 | 76 | 89 | 0.66 | 0.90 (0.85 to 0.94) |
| A1 | 50 | 84 | 81 | 0.64 | 0.89 (0.85 to 0.94) |
| A1.1 | 47 | 85 | 81 | 0.66 | 0.91 (0.86 to 0.95) |
| A2 | 8 | 88 | 91 | 0.79 | 0.87 (0.68 to 1.00) |
| A2.1 | 4 | 75 | 86 | 0.61 | 0.73 (0.29 to 1.00) |
| A2.3 | 3 | 100 | 99 | 0.99 | 1.00 (0.99 to 1.00) |
| Type B |  |  |  |  |  |
| B | 5 | 100 | 73 | 0.73 | 0.90 (0.80 to 1.00) |
| B1 | 5 | 100 | 73 | 0.73 | 0.90 (0.80 to 1.00) |
| B1.1 | 4 | 100 | 88 | 0.88 | 0.95 (0.90 to 1.00) |
| Type C |  |  |  |  |  |
| C | 12 | 92 | 81 | 0.73 | 0.90 (0.83 to 0.97) |
| C1 | 2 | 100 | 92 | 0.92 | 0.96 (0.88 to 1.00) |
| C1.1 | 2 | 100 | 86 | 0.86 | 0.93 (0.79 to 1.00) |
| C3 | 8 | 100 | 80 | 0.80 | 0.91 (0.85 to 0.97) |
| C3.2 | 6 | 100 | 92 | 0.92 | 0.95 (0.92 to 0.98) |

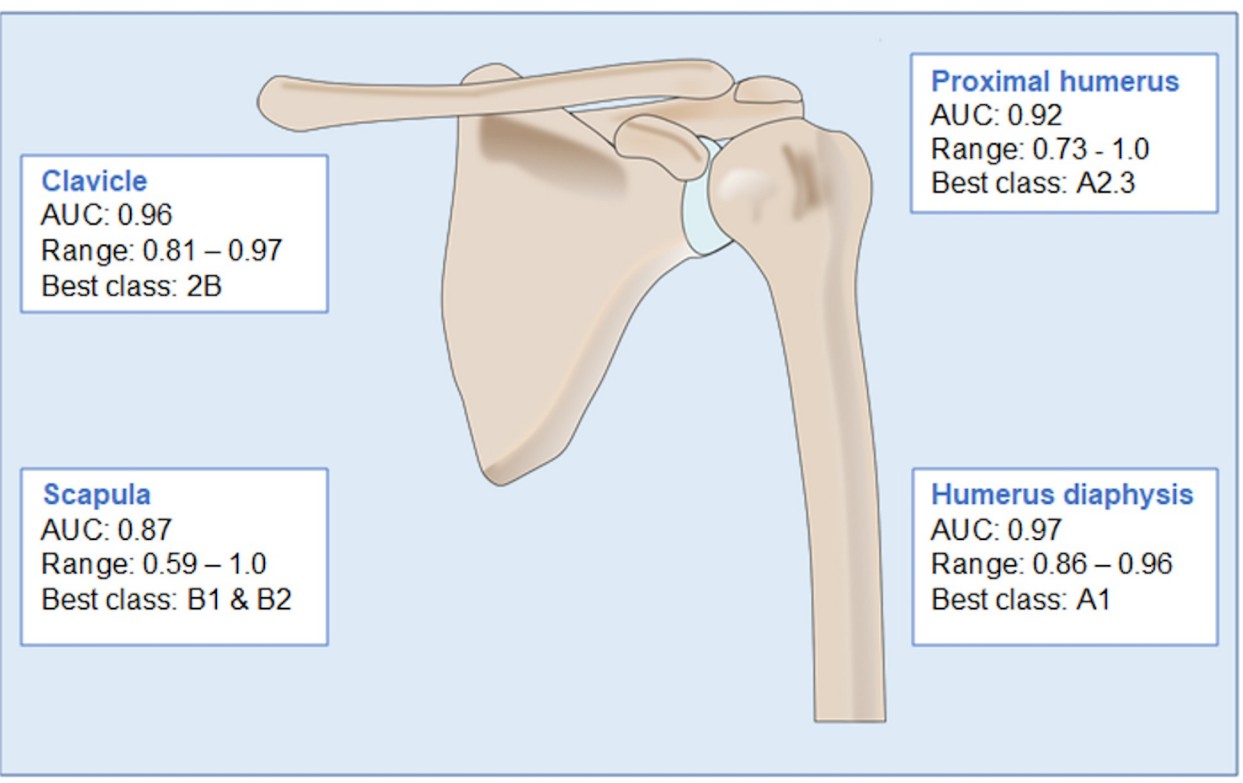

**Fig 2. Model performance for the different fractures, a summary.**

(C3.2), with AUC 0.95 (95% CI 0.92–0.98). Anatomical neck fracture with a multifragmentary metaphyseal fracture, diaphyseal extension class (subgroup C3.3), was not represented in the test data.

**Diaphyseal humerus fractures.** The training data included 216 diaphyseal humerus fractures, the validation data 39, and the test data 40. The overall precision for diaphyseal fractures was excellent with an AUC of 0.97 (95% CI 0.94 to 0.99). The range for AUC for fracture type and group was 0.88 to 0.97 (Table 3).

**Table 3. Diaphyseal humerus fractures.**

| | | Measurements | | | |
|---|---|---|---|---|---|
| | Cases (n = 406) | Sensitivity (%) | Specificity (%) | Youden's J | AUC (95% CI) |
| Type A | | | | | |
| A | 18 | 94 | 88 | 0.83 | 0.94 (0.91 to 0.98) |
| A1 | 4 | 100 | 92 | 0.92 | 0.96 (0.92 to 1.00) |
| A2 | 8 | 100 | 68 | 0.68 | 0.89 (0.80 to 0.98) |
| A3 | 6 | 100 | 85 | 0.85 | 0.95 (0.90 to 0.99) |
| Type B | | | | | |
| B | 14 | 93 | 86 | 0.79 | 0.93 (0.87 to 0.99) |
| B2 | 10 | 100 | 86 | 0.86 | 0.95 (0.91 to 0.98) |
| B3 | 3 | 100 | 92 | 0.92 | 0.95 (0.91 to 1.00) |
| Type C | | | | | |
| C | 8 | 88 | 90 | 0.77 | 0.91 (0.84 to 0.99) |
| C3 | 6 | 83 | 86 | 0.70 | 0.86 (0.71 to 1.02) |

**Table 4. Clavicle fractures.**

|  |  | Measurements | | | |
| --- | --- | --- | --- | --- | --- |
|  | Cases (n = 406) | Sensitivity (%) | Specificity (%) | Youden's J | AUC (95% CI) |
| Location 2 Diaphyseal segment | | | | | |
| 2 | 35 | 89 | 98 | 0.86 | 0.97 (0.94 to 1.00) |
| 2A | 16 | 81 | 88 | 0.70 | 0.87 (0.77 to 0.98) |
| 2B | 5 | 100 | 96 | 0.96 | 0.97 (0.95 to 0.99) |
| 2C | 13 | 100 | 88 | 0.88 | 0.98 (0.96 to 1.00) |
| Location 3 Lateral segment | | | | | |
| 3 | 15 | 93 | 70 | 0.63 | 0.90 (0.82 to 0.97) |
| 3A | 9 | 75 | 90 | 0.65 | 0.87 (0.76 to 0.99) |
| 3B | 5 | 80 | 99 | 0.79 | 0.81 (0.44 to 1.17) |

### Clavicle fractures

The training data included 749 clavicle fractures, the validation data 87, and the test data 51. The overall precision for clavicle fractures was excellent with an AUC of 0.96 (95% CI 0.92 to 0.99). The range for AUC for fracture type and group was 0.82 to 0.98 (Table 4).

### Scapula fractures

The training data included 243 scapula fractures, the validation data 87, and the test data 12. The overall precision for scapula fractures was good with an AUC of 0.87 (95% CI 0.92 to 0.99). The range for AUC for fracture type and group was 0.74 to 1.00 (Table 5).

### Other analyses

The interrater reliability was analysed using Cohen's Kappa and was 0.88 overall, for PHF 0.89, for diaphyseal fracture 0.88, for clavicle fractures 0.92, and for scapula fractures 0.71 [17].

## Discussion

In this paper, we present a deep learning model for fracture classification on shoulder radiographs. The overall AUC for fracture classification was 0.89, a good result by any standards.

**Table 5. Scapula fractures.**

|  |  | Measurements | | | |
| --- | --- | --- | --- | --- | --- |
|  | Cases (n = 406) | Sensitivity (%) | Specificity (%) | Youden's J | AUC (95% CI) |
| Location A Process | | | | | |
| A | 2 | 100 | 59 | 0.59 | 0.59* |
| Location B Body | | | | | |
| B | 5 | 100 | 100 | 1.00 | 1.00 (0.99 to 1.00) |
| B1 | 2 | 100 | 100 | 1.00 | 1.00 * |
| B2 | 3 | 100 | 99 | 0.99 | 1.00 (0.99 to 1.00) |
| Location F Glenoid Fossa | | | | | |
| F | 5 | 100 | 72 | 0.72 | 0.83 (0.66 to 1.00) |
| F1 | 5 | 100 | 68 | 0.68 | 0.83 (0.64 to 1.01) |
| F1.1 | 4 | 100 | 70 | 0.70 | 0.70 (0.65 to 0.75) |

*CI could not be calculated due to too few cases.

The results for PHF classification were even more impressive, with 12 of 16 classes achieving an AUC of 0.90 or greater.

To our knowledge, this is the first report to evaluate classification performance of a CNN classifier using the AO/OTA classification of PHF. Our CNN model demonstrated high classification accuracy for most fracture types, and our findings are in concordance with the few previous studies on the applications of AI networks as fracture classifiers [14,18–20]. The findings presented here further consolidate CNN as potent fracture classifiers in plain radiographs.

In addition, the CNN model had excellent overall AUC for diaphyseal humerus fractures (0.97) and for clavicle fractures (0.96), and good AUC for scapula fractures (0.87).

When compared to other studies on AI-assisted fracture classification, our study included a sample of 6,172 examinations in training, which includes 2–7 radiographs per examination, resulting in over 12,000 images. In comparison with other studies, our training set is twice the size of Urakawa and three times the size of Chung et al. [14,21].

## Strengths and limitations

A major strength of this study is the large study sample and the inclusion of multiple radiographs with different views from each examination. We tried to include as many radiographs as possible, and not excluding radiographs with suboptimal projection. In the clinical setting, radiograph quality is seldom perfect and depends on patient compliance. AI models for clinical use must be trained on data that represent the clinical reality to be useful. Furthermore, the study sample did not only include fracture radiographs. Including non-fracture radiographs is important for models that are aimed for use in the clinical setting where fractures will only be present in a minority of radiographs. A clinically useful AI model must be able to perform well despite the skewed distribution of fracture and fracture classes and be able to identify rare fracture classes. Our msodel performed well with AUC over 0.90 on several classes that only contained 3 cases.

One contributing factor to the high performance of our model might be the inclusion of several rather than singular radiographs. This further resembles clinical reality, where several radiographs are often assessed simultaneously in each examination. Another contributing factor to the model performance was not limiting training data to a specific projection, which provided the CNN with additional training data–a concept we considered beneficial. Additionally, the generally unselected nature of the sample, combined with a wide spectrum of injuries, further contributes to overall generalizability. The AO/OTA classification system is complex and may be more applied in research settings than in clinical settings. The use of AI-assisted classification models could potentially amplify the clinical adoption of the AO/OTA classification, while simultaneously enhancing clinicians' understanding of this classification system.

This study has several limitations. The ground truth labels in the training and validation datasets were provided by students with limited previous experience in plain shoulder radiograph assessment. This fundamental limitation permeates the study results because the CNN accuracy depends on the accuracy of the training data. A few measures were taken to reduce this influence. First, the validation data used to evaluate CNN classification performance was double audited. Second, students collaborated with a senior orthopaedic surgeon specialized in shoulder surgery, revisiting and reviewing complex and ambiguous cases. The test set was reviewed separately by two senior shoulder surgeons, and cases with discrepancies between the reviewers were handled through consensus sessions. During the consensus sessions, the surgeons were blinded to who had suggested the original classes.

Reliability of ground truth labels could have been further improved by having senior orthopaedic surgeons revisit and review all cases. Furthermore, having observers with previous

experience in radiographic assessment, such as radiologists or orthopaedic surgeons, provide the labels might have improved the accuracy of labels in the training data. In extension, such an approach would contribute to the overall reliability and generalizability of the study.

Several subgroups were not represented in the training data, and classification performance could not be evaluated in all AO/OTA PHF subgroups. By expanding the active learning selection bias, more subgroups could have been included in the training and validation datasets, enabling evaluation of classification performance in more of the AO/OTA PHF subgroups.

This is a single-centre study, and external validation of the CNN model should be done in the future before the model is considered for use in the clinical setting (Carmo et al. 2021).

This is the fifth study published on AI in orthopaedic radiographs by our group. The previous studies have demonstrated that AI can effectively utilise clinical radiographs and classify fractures with a classification system that was not primarily designed for AI use. These findings provide a promising indication that AI models may be implemented in clinical practice in the near future.

## Supporting information

**S1 Checklist. CLAIM: Checklist for Artificial Intelligence in medical imaging.**
(DOCX)

## Acknowledgments

We would like to express our gratitude to the Karolinska Institutet, for enabling collaboration between clinical researchers and students.

## Author Contributions

**Conceptualization:** Max Gordon.

**Data curation:** Max Gordon.

**Formal analysis:** Martin Magnéli, Max Gordon.

**Investigation:** Petter Ling, Jacob Gislén, Johan Fagrell, Yilmaz Demir, Erica Domeij Arverud, Kristofer Hallberg, Max Gordon.

**Methodology:** Björn Salomonsson, Max Gordon.

**Software:** Max Gordon.

**Supervision:** Björn Salomonsson.

**Validation:** Yilmaz Demir, Erica Domeij Arverud, Kristofer Hallberg.

**Visualization:** Martin Magnéli, Björn Salomonsson.

**Writing – original draft:** Martin Magnéli, Petter Ling, Jacob Gislén, Johan Fagrell.

**Writing – review & editing:** Yilmaz Demir, Erica Domeij Arverud, Kristofer Hallberg, Björn Salomonsson, Max Gordon.

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
