## [Decision Letter · Decision Letter 0]

12 May 2023

PONE-D-23-08539Deep learning classification of shoulder fractures and dislocations on plain radiographs of the humerus, scapula and claviclePLOS ONE

Dear Dr. Magneli,

Thank you for submitting your manuscript to PLOS ONE. After careful consideration, we feel that it has merit but does not fully meet PLOS ONE’s publication criteria as it currently stands. Therefore, we invite you to submit a revised version of the manuscript that addresses the points raised during the review process.

Both reviewers pointed out a great potential of this interesting study. However, a minor revision was recommended. Please, follow their instructions to improve the manuscript.

We look forward to receiving your revised manuscript.

Kind regards,

Hans-Peter Simmen, M.D., Professor of Surgery

Academic Editor

PLOS ONE

Journal Requirements:

3. We note that Figures 1 and 2 in your submission contain copyrighted images. All PLOS content is published under the Creative Commons Attribution License (CC BY 4.0), which means that the manuscript, images, and Supporting Information files will be freely available online, and any third party is permitted to access, download, copy, distribute, and use these materials in any way, even commercially, with proper attribution. For more information, see our copyright guidelines: http://journals.plos.org/plosone/s/licenses-and-copyright.

a. You may seek permission from the original copyright holder of Figures 1 and 2 to publish the content specifically under the CC BY 4.0 license. 

Reviewers' comments:

Reviewer's Responses to Questions

**Comments to the Author**

1. Is the manuscript technically sound, and do the data support the conclusions?

Reviewer #1: Yes

Reviewer #2: Yes

2. Has the statistical analysis been performed appropriately and rigorously? 

Reviewer #1: Yes

Reviewer #2: I Don't Know

3. Have the authors made all data underlying the findings in their manuscript fully available?

Reviewer #1: Yes

Reviewer #2: Yes

4. Is the manuscript presented in an intelligible fashion and written in standard English?

Reviewer #1: Yes

Reviewer #2: Yes

5. Review Comments to the Author

Reviewer #1: 1) General comment

The authors show in their study “Deep learning classification of shoulder and dislocations on plain radiographs of the humerus, scapula and clavicle” that their deep learning model based on a convolutional neural network (CNN) perfom promising results in fracture-classification and degenerative conditions.

I guess the artificial intelligence (AI) is coming a key player in prevention, diagnosis, treatment, patient monitoring and data management in medicine. Therefore machine learning/neuronal networks and deep neuronal networks are very important methodical approaches in the future. The applied CNN is one type of deep neuronal networks which based on different hidden layers with billions of nodes and is a further development of neuronal networks. In this way CNN are used in recognizing of images.

All in all, it is an interesting study about fracture classification based on artificial intelligence using a CNN. An overall area under the curve (AUC) has a good performance for fracture classification related to proximal and diaphyseal humerus fractures. Additionally the authors show good to excellent AUC for clavicule, and scapula fractures.

Overall this study is a good example as the algorhythms of deep neuronal networks can work with unstructured data without processing to structured data. The algorhythms are extracting itself the essential characteristics to differentiate the data. Finally, this CNN model recognize patterns and interpret the variations.

The demonstrated application of AI in fracture classification is a very interesting example in medicine. And the possible applications will probably expand to other medical fields.

2) Specific suggestions

Abstract

Line 32: ……………………results for both fractures adn degenerative conditions

The study conducted by the authors focuses on applying CNN to fracture-classification and not to degenerative bone changes. So I would omit the remarks about degenerative conditions of bone.

Introduction

Line 59:……………………anterior dislocation of the shoulder is a common …..

The topic of shoulder dislocation injuries does not fit into the topic of fraucture classifications genereated using deep neuronal networks (DNN). In addition, the current classification of shoulder dislocation injuries often does not contribute to the indication for surgery and they are discussed controversially.

Material and Methods

Line 129:………………………dislocations of the acromioclavicular joint were classified

The classification of dislocation injuries of acromioclavicular joints is missing from Table 1. But I would omit it altogether anyway as this entity does not contribute anything significant to the main topic.

Results

Line 225/226…………………………The training data included 432 dislocations…….

The entity of acromioclavicular joint injuries particularly Type III injuries (Rockwood) , is not sufficiently present in the test group for an adequate assessment of CNN

For the reasons already mentioned above and clarity, I would omit this

Discussion

Line 270/271………………..AI might give birth to new ways of classifiyng fractures, ideally moving from a system where the classification describes the look of the fracture to a system based……

This sentence is misleading. A diagnostic tool should generate a diagnosis that can be used a decision-making aid in order to make a possible indication for surgery. This means that you don`t need AI to record how the fracture behaves under a specific type of treatment.

Reviewer #2: The authors present a study on the learning ability of artificial intelligence to assess radiographs of the proximal humerus. The learned network showed excellent accuracy in the classification of fractures according to AO/OTA.

The study reads smoothly, is written in standard English, and has potential great value in the future of medicine.

AI and neuronal networks are not my area of specialization, comments regarding the method section from this reviewer are therefore not free of doubt. Nevertheless, the statistics used are common in clinical practice to provide valuable information about the value of an investigation.

Comments:

Line 59: The upbringing of shoulder dislocations comes quite surprisingly at this point. Until here it was not clear, that dislocations should be evaluated as well. Therefore, it is recommended to elaborate this paragraph further.

Line 270 ff: This sentence is not supported by the data provided. The study is all about the in orthopedic surgery very standard AO/OTA classification, which by the way is used, at least from my side, even in everyday practice. New ways of classification of fractures is something completely different compared to training AI with an already existing fracture classification system. And how should such a new system change existing treatment. This brings the discussion to a level, were the data can not support the authors in their arguing. Nevertheless I would love to be proven wrong. But in this case much further elaboration on this is needed.

6. PLOS authors have the option to publish the peer review history of their article (what does this mean?). If published, this will include your full peer review and any attached files.

Reviewer #1: No

Reviewer #2: **Yes: **Samuel Haupt

---

## [Author Response · Author response to Decision Letter 0]

3 Jul 2023

Dear Professor Simmen,

I appreciate the opportunity to revise our manuscript entitled "Deep learning classification of shoulder fractures and dislocations on plain radiographs of the humerus, scapula and clavicle" for potential publication in PLOS ONE. My team and I are grateful for the reviewers' constructive feedback, and we believe that their comments have significantly improved our paper. We have revised the manuscript addressing each point raised by the reviewers and have provided a detailed response to the comments below.

**Reviewer 1 Comments**

1) General comment

The authors show in their study “Deep learning classification of shoulder and dislocations on plain radiographs of the humerus, scapula and clavicle” that their deep learning model based on a convolutional neural network (CNN) perfom promising results in fracture-classification and degenerative conditions.

I guess the artificial intelligence (AI) is coming a key player in prevention, diagnosis, treatment, patient monitoring and data management in medicine. Therefore machine learning/neuronal networks and deep neuronal networks are very important methodical approaches in the future. The applied CNN is one type of deep neuronal networks which based on different hidden layers with billions of nodes and is a further development of neuronal networks. In this way CNN are used in recognizing of images.

All in all, it is an interesting study about fracture classification based on artificial intelligence using a CNN. An overall area under the curve (AUC) has a good performance for fracture classification related to proximal and diaphyseal humerus fractures. Additionally the authors show good to excellent AUC for clavicule, and scapula fractures.

Overall this study is a good example as the algorhythms of deep neuronal networks can work with unstructured data without processing to structured data. The algorhythms are extracting itself the essential characteristics to differentiate the data. Finally, this CNN model recognize patterns and interpret the variations.

The demonstrated application of AI in fracture classification is a very interesting example in medicine. And the possible applications will probably expand to other medical fields.

Response: Thank you for your thoughtful and thorough review of our study. We appreciate your recognition of the potential of AI and our work in the field of fracture classification using CNNs. We are encouraged by your positive feedback and will continue to explore the potential applications of AI in medical fields.

1) "The study conducted by the authors focuses on applying CNN to fracture-classification and not to degenerative bone changes. So I would omit the remarks about degenerative conditions of bone."

Response: We appreciate your suggestion and agree that this might confuse the reader. To better represent our study focus, removed references to other labelled findings such as degenerative conditions. The removed sections were found on lines: 33, 97-98 in the first submission. We have removed reference number 13 on shoulder dislocations and updated the reference list, so it excludes this reference.

2) "The topic of shoulder dislocation injuries does not fit into the topic of fracture classifications generated using deep neuronal networks (DNN)."

Response: We understand your concern. We aimed to demonstrate the versatility of our model, including its application to dislocation injuries. However, to maintain focus on fracture classification, we have removed all sections on anterior shoulder dislocations, and removed dislocations from the title. The removed sections were found on lines: 59-60, 70, 128-130, 225-228, and removed reference number 13 from the first submitted manuscript.

3) "The classification of dislocation injuries of acromioclavicular joints is missing from Table 1. But I would omit it altogether anyway as this entity does not contribute anything significant to the main topic."

Response: We apologize for the oversight. However, following your valuable suggestion, we have now omitted dislocation injuries from our study to maintain focus on fracture classifications.

4) "This sentence is misleading. A diagnostic tool should generate a diagnosis that can be used a decision-making aid in order to make a possible indication for surgery."

Response: We agree with your comment. We did not mean to suggest that AI would replace clinical decision-making. Rather, we wanted to highlight that AI could assist in classifying fractures more objectively, potentially leading to more uniform treatment strategies. We have revised the sentence to avoid any confusion. The revised text is found on line xxx

**Reviewer 2 Comments**

1) "The upbringing of shoulder dislocations comes quite surprisingly at this point."

Response: We agree with your feedback and have now revised the introduction to provide a clearer picture of our study focus on fracture classifications, and removed all sections on dislocations (please see response to reviewer 1).

2) "This sentence is not supported by the data provided."

Response: We appreciate your critique and agrees with you that statement was not supported by the data. We have now revised the discussion to more accurately reflect the findings of our study. The revised text is found on lines 277-280.

**Journal Requirements**

1) We have ensured that our manuscript now follows PLOS ONE's style requirements.

2) We have updated the Data Availability statement to specify where the minimal dataset can be found. We will also include URLs or DOIs for the data in the cover letter. 

3) We have obtained written permission from the copyright holder to publish Figures 1 and 2 under the CC BY 4.0 license. This documentation will be uploaded as an "Other" file with our submission.

4) We have reviewed our reference list and confirmed that it is complete and correct. 

We sincerely thank you for considering our manuscript for publication in PLOS ONE. We believe the revised manuscript has addressed the concerns raised and hope that it is now acceptable for publication.

Best Regards,

Martin Magnéli, MD, PhD

---

## [Editor Report · Decision Letter 1]

27 Jul 2023

Deep learning classification of shoulder fractures on plain radiographs of the humerus, scapula and clavicle

PONE-D-23-08539R1

Dear Dr. Magneli,

We’re pleased to inform you that your manuscript has been judged scientifically suitable for publication and will be formally accepted for publication once it meets all outstanding technical requirements.

Kind regards,

Hans-Peter Simmen, M.D., Professor of Surgery

Academic Editor

PLOS ONE
---

## [Editor Report · Acceptance letter]

22 Aug 2023

PONE-D-23-08539R1 

Deep learning classification of shoulder fractures on plain radiographs of the humerus, scapula and clavicle 

Dear Dr. Magneli:

I'm pleased to inform you that your manuscript has been deemed suitable for publication in PLOS ONE. Congratulations! Your manuscript is now with our production department. 

Kind regards, 

on behalf of

Dr. Hans-Peter Simmen 

Academic Editor

PLOS ONE